# TT-NF: Tensor Train Neural Fields

## Abstract

Learning neural fields has been an active topic in deep learning research, focusing, among other issues, on finding more compact and easy-to-fit representations. In this paper, we introduce a novel low-rank representation termed Tensor Train Neural Fields (TT-NF) for learning neural fields on dense regular grids and efficient methods for sampling from them. Our representation is a TT parameterization of the neural field, trained with backpropagation to minimize a non-convex objective. We analyze the effect of low-rank compression on the downstream task quality metrics in two settings. First, we demonstrate the efficiency of our method in a sandbox task of tensor denoising, which admits comparison with SVD-based schemes designed to minimize reconstruction error. Furthermore, we apply the proposed approach to Neural Radiance Fields, where the low-rank structure of the field corresponding to the best quality can be discovered only through learning.

## 1 Introduction

Following the growing interest in deep neural networks, learning neural fields has become a promising research direction in areas concerned with structured representations. However, precision is usually at odds with the computational complexity of these representations, which makes training them and sampling from them a challenge. In this paper, we investigate interpretable low-rank neural fields defined on dense regular grids and efficient methods for learning them. Since, in extreme cases, the dimensionality of such fields can exceed the memory size of a typical computer by several orders of magnitude, we look at the problem of learning such fields from the angle of stochastic methods.

Tensor decompositions have become a ubiquitous tool for dealing with structured sparsity of intractable volumes of data. Within the large family of tensor decompositions, we focus on the Tensor Train (TT) (Oseledets, 2011), also known as the Matrix Product State in physics. TT is notable for its high-capacity representation, efficient algebraic operations in the low-rank space, and support of SVD-based algorithms for data approximation. As such, we consider TT-SVD (Oseledets, 2011) and TT-cross (Oseledets & Tyrtyshnikov, 2010) methods for obtaining a low-rank representation of the full tensor. While TT-SVD requires access to the full tensor at once (which might already be problematic in specific scenarios), TT-cross requires access to data through a black-box function, computing (or looking up) elements by their coordinates on demand. Both methods operate under the assumption of noise-free data and are not guaranteed to output sufficiently good approximations in the presence of noise.

While noise in observations is challenging for SVD-based schemes and requires devising tailored approaches to different noise types and magnitude (Zhou et al., 2022), exploiting the low-rank structure of the field driven by data is even more challenging (Novikov et al., 2014; Boyko et al., 2020) and typically resorts to the paradigm of data updates through algebraic operations on TT.

In this work, we take a step back and leverage the modern deep learning paradigm to parameterize neural fields as TT, coined TT-NF. Through deep learning tooling with support for automatic differentiation and our novel sampling methods, we obtain mini-batches of samples from the parameterized neural field and perform optimization of a non-convex objective defined by a downstream task. The optimization comprises the computation of parameter gradients with backpropagation and parameter updates with any suitable technique, such as SGD.

We analyze TT-NF and several sampling techniques on a range of problem sizes and provide reference charts for choosing a sampling method based on memory and computational constraints. Next, we define a synthetic task of low-rank tensor denoising and demonstrate the superiority of the proposed

optimization scheme over several SVD-based schemes. Finally, we consider the formulation of Neural Radiance Fields (NeRF) introduced in Mildenhall et al. (2020), and propose a simple modification to TT-NF, termed QTT-NF, for dealing with hierarchical spaces.

Our contributions in this paper:

1. TT-NF – compressed low-rank neural field representation defined on a dense grid;
2. QTT-NF – a modification of TT-NF for learning neural fields defined on hierarchical spaces, such as 3D voxel grids seen in neural rendering;
3. Efficient algorithms for sampling from (Q)TT-NF and learning it from samples, designed for deep learning tooling.

The rest of the paper is organized as follows: Sec. 2 discusses the related work; Sec. 3 introduces notations from the relevant domains; Sec. 4 presents the proposed contributions; Sec. 5 demonstrates the practical use of the proposed methods; Sec. 6 concludes the paper. Many relevant details pertaining to our method, experiments, and extra discussion can be found in Appendix sections A, B, and C.

## 2 RELATED WORK

**Tensor Decompositions** Higher-order tensor decompositions have been found helpful for several data-based problems, as detailed by Kolda & Bader (2009). Oseledets (2011) introduced the Tensor Train (TT) decomposition, which offers a compressed low-rank tensor approximation that is stable and fast. The TT decomposition has also been used to approximate tensors with linear complexity in their dimensionality via the TT-cross approximation (Oseledets & Tyrtyshnikov, 2010). With the rise of deep learning, tensor-based methods have been integrated into neural networks, e.g., Usvyatsov et al. (2021) explored the use of TT-cross approximation for gradient selection in learning representations. We review tensor-based methods for network compression in the next paragraph and refer the reader to Panagakis et al. (2021) for a detailed overview of similar works. On the software side, along with general deep learning frameworks (Paszke et al., 2019; Abadi et al., 2015), several tensor-centric frameworks have emerged (Kossaifi et al., 2019b; Usvyatsov et al., 2022; Novikov et al., 2020).

**Neural Network Compression with Tensors** Low-rank bases were utilized by Jaderberg et al. (2014) to approximate convolutional filters and drastically speed up inference via separating filter depth from spatial dimensions. Lebedev et al. (2014) applied a low-rank decomposition on all 4 dimensions of the standard convolutional kernel tensors. Subsequent works employed more general tensor decompositions, notably the TT decomposition, to massively compress fully connected layers (Novikov et al., 2015) or both fully connected and convolutional layers (Garipov et al., 2016), with minor accuracy losses. Kossaifi et al. (2019a) applied the higher-order tensor factorization to the entire network instead of separately to individual layers. In a similar vein, Li et al. (2019); Obukhov et al. (2020); Kanakis et al. (2020) propose to learn a basis and coefficients of each layer, thus enabling disentangled compression and multitask learning. While most of the aforementioned methods examine general convolutional networks, we focus specifically on compressing neural fields.

**Tensor Decompositions in 3D Representations** Early approaches in 3D volume representation used global low-rank tensor bases for reconstruction at multiple scales and resolutions (Suter et al., 2013). A review of compact representations and sampling techniques for compressed direct volume rendering (DVR) is made by Balsa Rodríguez et al. (2014). Ballester-Ripoll et al. (2015) analyzed multiple tensor approximation models regarding volume visualization. More recently, compression of 3D volumes over regular grids was addressed with a general higher-order singular value decomposition (SVD) by Ballester-Ripoll et al. (2019). The TT decomposition has also been used by Boyko et al. (2020) to compress 3D scenes that are represented by volumetric distance functions. We review neural-field-based methods separately in the next paragraph.

**Neural Fields** Neural fields as implicit scene representations for geometry and radiance have recently attracted intense research activity, especially in the context of 3D. The application of neural fields to image compression is studied by Strümpler et al. (2021), who employ meta-learned representations that increase efficiency in training. The usual volumetric type of representation is replaced by a surface-based one by Zhang et al. (2021a), who learned bidirectional reflectance

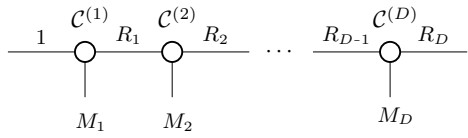 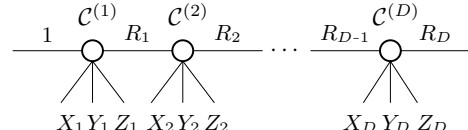

Figure 1: Tensor diagram of the (Block) Tensor Train decomposition (Dolgov et al., 2014). The low-rank tensor of shape $M_1 \times ... \times M_D \times R_D$ is represented as a product of $D$ TT-cores $\mathcal{C}^{(i)}$, each being a tensor of shape $R_{i-1} \times M_i \times R_i$. The TT-rank $(1, R_1, ..., R_D)$ defines the degree of approximation. The case of $R_D = 1$ corresponds to TT decomposition Oseledets (2011).

Figure 2: Tensor diagram of a Quantized Tensor Train decomposition (Khoromskij, 2009) of a 3D voxel grid of shape $X \times Y \times Z$ and $R_D$ values in each voxel. All three dimensions admit factorization into $D$ levels of hierarchy (e.g., $X = X_1 X_2 \cdots X_D$). We group factors by levels into tuples $(X_i, Y_i, Z_i)$, and introduce the low-rank bonds between the levels of the hierarchy.

distribution functions that enable novel view synthesis for unconstrained real-world scenes, and Zhang et al. (2021b), who designed a signed distance field to model the scene geometry. For a comprehensive overview of neural fields in visual computing, we refer the reader to Xie et al. (2022).

**Neural Radiance Fields**    The seminal paper of Mildenhall et al. (2020) introduced the concept of a NeRF as an end-to-end implicit differentiable function that maps viewpoints to scene renderings and showed that it could be effectively optimized. The area advances quickly; we further discuss works focusing on representation sparsity and efficiency. Yu et al. (2022) employed a sparse 3D-grid representation based on spherical harmonics and regularization for novel view synthesis, alleviating the need for neural components. Müller et al. (2022) improved the training efficiency by using a hash encoding for the input feature vectors, which allows using a much smaller network. Reiser et al. (2021) improved the rendering efficiency by assigning different parts of the scene to different small networks. Sun et al. (2022) achieved faster convergence to optimal solutions for volumetric representations by using a post-activation interpolation and guiding the optimization via prior knowledge of the problem. Chen et al. (2022) implemented low-rank constraints in 3D on the voxel grid parameterization and achieved impressive results in the NeRF setting.

## 3    NOTATION

**Tensor Diagram Notation**    Diagrams are an efficient visualization tool for interactions between tensors. A tensor is drawn as a node with the number of legs matching its number of dimensions. For example, a matrix $W \in \mathbb{R}^{m \times n}$ is drawn as ─o─, and a vector $x \in \mathbb{R}^n$ looks as ─o. Their product $Wx$ looks as a connection along the dimension being eliminated as a result of the operation: ─o─o. A diagram of nodes with their connections reflects what is called a "tensor network".

**Tensor Contraction** computes a product of the entire tensor network. The result of this operation is a single tensor with dimensions corresponding to free legs inside the tensor network.

**Tensor Train Decomposition**    The TT format represents a $D$-dimensional array (tensor) $\mathcal{A} \in \mathbb{R}^{M_1 \times M_2 \times \cdots \times M_D}$ with modes $M_i$ as a product of $D$ three-dimensional core tensors $\mathcal{C}^{(i)} \in \mathbb{R}^{R_{i-1} \times M_i \times R_i}$, called TT-cores (see Fig. 1). The tuple $(R_0, R_1, ..., R_D)$ is called a TT-rank of the decomposition; it defines the degree of approximation of $\mathcal{A}$. By convention (Oseledets, 2011), $R_0 = R_D = 1$. $R_{\max} = \max(R_0, ..., R_D)$ is also called the rank of the decomposition. An element of $A$ at indices $(i_1, ..., i_D)$ can be computed as follows:

$$A_{i_1,...,i_D} = \sum_{\beta_1,...,\beta_{D-1}=1}^{R_1,..,R_{D-1}} \mathcal{C}^{(1)}_{1,i_1,\beta_1} \cdot \mathcal{C}^{(2)}_{\beta_1,i_2,\beta_2} \cdots \mathcal{C}^{(D-1)}_{\beta_{D-2},i_{D-1},\beta_{D-1}} \cdot \mathcal{C}^{(D)}_{\beta_{D-1},i_D,:} \tag{1}$$

Dolgov et al. (2014) introduced Block TT, which attaches a "block" dimension in place of the last rank $R_D$. The difference between the two is subtle, as both formats can be converted to each other. However, Block TT is more suitable for describing multi-valued neural fields. We thus assign a special meaning to $R_D$ – it will signify the "payload" dimension of our neural field. We will omit the word "block" in the remaining text and always assume Block TT. Refer to Fig. 1 for the diagram.

**QTT**   Mode Quantization refers to the introduction of artificial dimensions into the represented tensor. For example, a 3D tensor of function values on the lattice $16 \times 16 \times 16$ can be represented as a tensor of shape $(2_1 \times 2_2 \times 2_3 \times 2_4) \times (2_1 \times 2_2 \times 2_3 \times 2_4) \times (2_1 \times 2_2 \times 2_3 \times 2_4)$. Here color-coded factors denote one of the axes $X$, $Y$, and $Z$ they explain, subscripts denote the artificially-introduced levels of hierarchy, $\times$ delimits represented dimensions, $\cdot$ denotes merged (flattened) dimensions, and parentheses are for visual convenience. This 12-dimensional cube with side 2 can then be represented using TT as in Fig. 1. However, when the number of introduced levels of hierarchy is the same between modes of the original tensor (four in the given example), it is often beneficial (Oseledets, 2009) to introduce low-rank structure between levels of hierarchy, rather than individual factors. The resulting low-rank representation describes a tensor of shape $8 \times 8 \times 8 \times 8$ with the following factorization pattern called QTT: $(2_1 \cdot 2_1 \cdot 2_1) \times (2_2 \cdot 2_2 \cdot 2_2) \times (2_3 \cdot 2_3 \cdot 2_3) \times (2_4 \cdot 2_4 \cdot 2_4)$. Notably, such permutation of dimension factors corresponds to 3D space traversal using Morton code (Morton, 1966), also known as Z-order. Connections with octrees used in rendering can also be drawn.

We define our neural field on a 3D voxel grid as shown in Fig. 2. Dimensions of the voxel grid are chosen equal to $2^D$, resulting in a hierarchy of $D$ levels. Together with the Block structure discussed above and QTT, we arrive at the proposed representation, which has $D$ cores with all modes equal to eight, and the payload dimension corresponding to the number of values to store in each voxel.

**Tensor Decomposition**   The term may refer either to the decomposition scheme (e.g., as in the figures above) or the process of obtaining values of the decomposition (e.g., TT-cores $\mathcal{C}$) from elements of the full tensor $\mathcal{A}$. While SVD-based schemes employ the latter meaning, we focus on the former by parameterizing decomposition given its configuration, which includes modes and TT-rank.

**Sampling** refers to obtaining elements of the full tensor from its decomposition at a given list of indices. This operation could be done in several ways, the simplest being Tensor Contraction followed by subsampling at the required indices. When the contraction is undesired or intractable, an alternative way is evaluating full tensor elements through the decomposition equation, such as Eq. 1. While mathematically straightforward, the subtleties of the chosen sampling algorithm result in a large variance in efficiency when used within the optimization loop due to the need for automatic differentiation, as we discuss in Sec. 4.2.

## 4   METHOD

We introduce TT-NF as a parameterization of the TT decomposition discussed in Sec. 3 and use it within a deep learning framework with automatic differentiation support. This paradigm change differs from previous methods for obtaining tensor decompositions, relying on matrix decompositions (SVD, QR, *etc.*) and algebraic operations in the TT format. Each such scheme (collectively called "SVD-based") comes with its own set of limitations, summarized in Tab. 1. TT-SVD (Oseledets, 2011) assumes access to all elements of the full tensor $\mathcal{A}$ in memory, which may be intractable in specific large-scale scenarios, and does not support noise in observations. TT-Cross (Oseledets & Tyrtyshnikov, 2010) accepts a black-box function for computing elements of the full tensor on-demand, which suits

Table 1: Comparison of methods for obtaining a TT decomposition from observations: TT-SVD (Oseledets, 2011), TT-Cross (Oseledets & Tyrtyshnikov, 2010), TT-OI (Zhou et al., 2022), TT-NF.

| Method | Observation access pattern | Noise in observations |
|---|---|---|
| TT-SVD | Full tensor | Not supported |
| TT-Cross | On-demand, pattern defined by dimensions and TT-rank | Not supported |
| TT-OI | Full tensor | Sub-gaussian |
| TT-NF (our) | On-demand, flexible batch size and access pattern | Any supported by the choice of the loss function |

large-scale problems, but lacks the flexibility in choosing the number of samples through which update is performed (batch size); this number is defined purely by the configuration of TT decomposition (dimensions and TT-rank). Likewise, it does not support noise in observations. TT-OI (Zhou et al., 2022) improves upon TT-SVD and supports zero-mean independent sub-gaussian noise in observations but inherits scalability issues.

Our method stands out due to its flexibility of optimization parameters choice (e.g., batch size) and resilience to various types of noise in observations, controlled through the choice of the loss function.

### 4.1 INITIALIZATION OF TT-NF

Given the field's dimensions, we first choose its TT-rank. For that, we choose the value of $R_{\max}$ and set TT-rank to the maximum possible values according to Oseledets (2011), not exceeding $R_{\max}$.

Following the best practices in the deep learning literature, we initialize parameters of TT-NF from scratch using the normal distribution with scale $\hat{\sigma}$ computed such that the full tensor elements computed using Eq. 1 have a pre-defined scale $\sigma$, as shown in Eq. 2:

$$\hat{\sigma} = \exp\left(\frac{1}{2D}\left(2\log\sigma - \sum_{i=1}^{D}\log R_i\right)\right). \tag{2}$$

Alternatively, in the presence of access to full tensor elements, parameters can be initialized using the output of any of the SVD-based schemes, leading to faster convergence.

### 4.2 SAMPLING FROM TT-NF

Obtaining samples from TT-NF is mathematically straightforward using Eq. 1. In deep learning frameworks, one way to obtain a batch of $B$ samples at indices $\left(\left(i_1^{(1)}...i_D^{(1)}\right),...,\left(i_1^{(B)}...i_D^{(B)}\right)\right)$ amounts to using `index_select` operation on each TT-core $\mathcal{C}_i$ along mode $M_i$ to obtain batches of core slices (matrices) of shapes $B \times R_{i-1} \times R_i$, and then applying `bmm` (batched matrix multiplication) operation to them. This sampling method (aliased **v1**) leads to space complexity of $O(BDR_{\max}^2)$ and quickly becomes unusable as slices of parameterization are replicated in memory for each sample. Moreover, the scaling issue gets worse as we require keeping all intermediate computations after each `bmm` operation and allocating memory for gradients to enable automatic differentiation.

The prevention of model parameter replication enabled the scaling of modern neural networks with millions of parameters, which are trained using minibatches of thousands of samples. The cornerstone of efficient scaling is a set of specialized layers (e.g., `Linear`), which accept a batch of inputs and compute mappings using only one instance of parameters. During the backward pass, gradients w.r.t. parameters are accumulated from samples with predictable, constant space scaling.

With that in mind, we bootstrap our efficient sampling method (aliased **v2**) by leveraging `Linear` layer functionality. Given a batch of indices as above, we start by taking $\left(i_1^{(1)}...i_D^{(1)}\right)$ and produce a batch of intermediates $v$ of shape $B \times R_1$ (the dimension corresponding to $R_0 = 1$ is ignored). For each subsequent TT-core $\mathcal{C}_i, i = \overline{2, D}$, we split the inputs $v$ according to which $M_i$ slices of $\mathcal{C}_i$ will be used to perform vector-matrix multiplication of each sample. Because `Linear` layers require samples in the minibatch to be packed densely, we perform a permutation $\pi$ of $v$ to align samples in the minibatch before multiplying them with the respective weight matrices. We additionally maintain the inverse permutation $\sigma$ to restore the order of samples after processing $v$ with the last TT-core. The output of each step has the shape $B \times R_i$ compatible with the input to the next step until reaching the last step, where it is of the shape $B \times R_D$. Alg. 1 outlines the details of the algorithm.

The resulting space complexity of **v2** sampling is reduced to $O(BDR_{\max})$, with the absorbed scaling factor just $2\times$ that of the **v1** method due to permutations. The memory footprint of **v2** is roughly $R_{\max}$ times smaller than **v1**, which enables practical use for TT-NF optimization as the rank increases.

Fig. 3 provides a reference chart for choosing an optimal sampling scheme based on an uncompressed field size of $2^{30}$, batch size, $R_{\max}$, space, and time constraints. Refer to additional charts for different field and batch sizes in the Appendix, Figs. 7, 8. As can be seen, **v2** consistently outperforms **v1** in memory requirements and sampling through tensor contraction followed by indexing in both memory and FLOPs, given the batch sizes specified in the plot. Additionally, we introduce a reduced parameterization and an associated **v3** sampling method in Sec. A.1.

**Algorithm 1** Memory-Efficient Sampling from TT-NF for Deep Learning Frameworks. Automatic differentiation paths are highlighted in blue. Refer to Sec. 4.2 for more details.

**Require:**
$D$ - number of tensor dimensions,
$B$ - number of samples,
$(1, R_1, ..., R_D)$ - TT-rank,
$(M_1, ..., M_D)$ - TT-modes,
$(\mathcal{C}^{(1)}, ..., \mathcal{C}^{(D)})$ - TT-cores representing $\mathcal{A}$,
$\left(\left(i_1^{(1)}...i_D^{(1)}\right), ..., \left(i_1^{(B)}...i_D^{(B)}\right)\right)$ - indices.

**Ensure:**
$v = \left(\mathcal{A}_{i_1^{(1)},...,i_D^{(1)}}, ..., \mathcal{A}_{i_1^{(B)},...,i_D^{(B)}}\right)$ - samples
without computing the whole $\mathcal{A}$.

1: $\pi \leftarrow (1, ..., B)$  ▷ forward permutation
2: $\sigma \leftarrow (1, ..., B)$  ▷ inverse permutation
3: $v \leftarrow \mathcal{C}^{(1)}_{1,i_1,:}$  ▷ $B \times R_1$
4: **for** $k \leftarrow 2$ to $D$ **do**
5:   $i_k \leftarrow \pi(i_k)$  ▷ align mode indices
6:   $v, \pi_k \leftarrow$ BIMVP$(\mathcal{C}^{(k)}, i_k, v)$  ▷ $B \times R_k$
7:   $\sigma_k \leftarrow \pi_k^{-1}$  ▷ invert $k^{\text{th}}$ permutation
8:   $\pi \leftarrow \pi_k(\pi)$  ▷ update forward perm.
9:   $\sigma \leftarrow \sigma(\sigma_k)$  ▷ update inverse perm.
10: **end for**
11: $v \leftarrow \sigma(v)$  ▷ recover samples order
12: **return** $v$  ▷ $B \times R_D$

**Algorithm 2** Batched-Indexed Matrix-Vector Permuted Product (BIMVP) for Deep Learning Frameworks, referenced in Alg. 1. Automatic differentiation paths are highlighted in blue.

**Require:**
$\mathcal{C}$ - TT-core of shape $R_l \times M \times R_r$,
$i$ - $B$ indices in $[1, M]$,
$v$ - batch of vectors of shape $B \times R_l$.

**Ensure:**
$\pi(v_{i_1}\mathcal{C}_{:,i_1,:}, ..., v_{i_B}\mathcal{C}_{:,i_B,:})$ - permuted output,
$\pi$ - permutation.

1: $\pi \leftarrow$ argsort$(i)$  ▷ compute permutation
2: $b_1, ..., b_M \leftarrow$ unique$(i)$  ▷ count $M$ unique
3: $v \leftarrow \pi(v)$  ▷ group vectors by matrices
4: $v_1, ..., v_M \leftarrow$ split$(v, b)$  ▷ split groups
5: **parfor** $m \leftarrow 1$ to $M$ **do**
6:   ▷ Linear layer function without bias
7:   $v_m \leftarrow$ linear$(v_m, \mathcal{C}^{\top}_{:,m,:})$  ▷ $b_m \times R_r$
8: **end parfor**
9: $v \leftarrow (v_1, ..., v_M)$  ▷ concatenate vectors
10: **return** $v, \pi$  ▷ $B \times R_r$

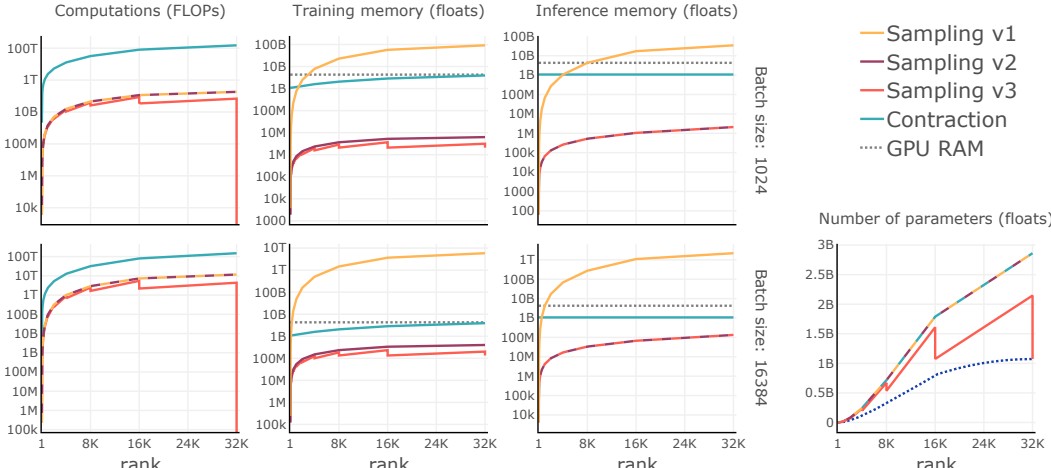

Figure 3: Space-time complexity of sampling from TT-NF of size $2^{30}$ with various methods, batch sizes, and ranks. We compare three sampling schemes discussed in Sec. 4, as well as the traditional tensor contraction scheme. As seen in the plots, **v2** (Sec. 4.2) scheme requires orders of magnitude fewer floating point operations (FLOPs) and memory than the contraction scheme and outperforms naive **v1** sampling in memory requirements. Additionally, **v3** (Sec. A.1) offers extra speedup on top of **v2** without loss of representation capacity, bringing its parameterization closer to the theoretical number of degrees of freedom of the tensor train manifold. Lower is better. Best viewed in color.

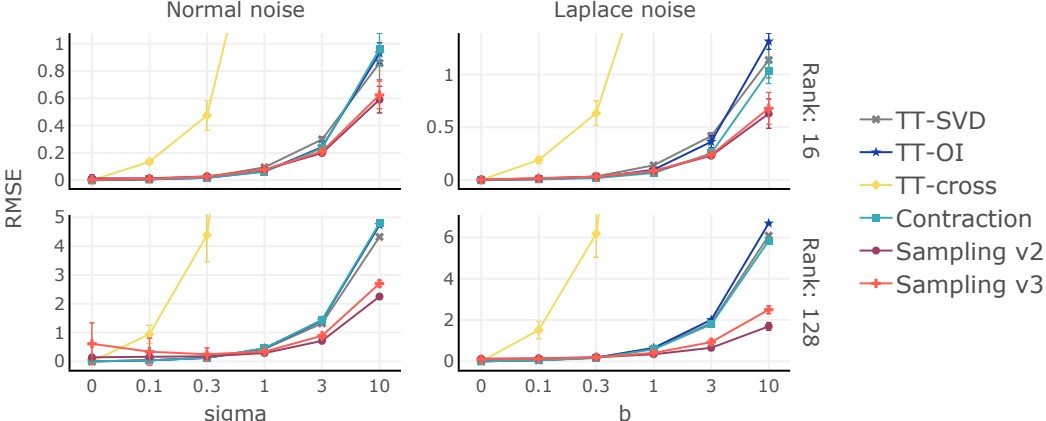

Figure 4: The resilience of various tensor regression methods of size $2^{20}$ to additive noise. We consider zero-mean Normal and Laplace noise with varying scales added to random ground truth with unit variance and known TT decomposition. We report root mean squared errors (RMSE) of tensors regressed by each method. Our sampling methods (**v2** and **v3**) outperform other methods in noisy settings, including those specifically designed to work with noisy observations (TT-OI by Zhou et al. (2022)) and those computing a fraction of elements at each optimization step (TT-cross by Oseledets & Tyrtyshnikov (2010)). See discussion in Sec. 5.1. Lower is better. Best viewed in color.

## 5 EXPERIMENTS

### 5.1 TENSOR DENOISING

To compare TT-NF with SVD-based schemes, we use a synthetic task of tensor denoising: given a noisy observation $\mathcal{Y} \in \mathbb{R}^{M_1 \times \dots \times M_D}$ of a tensor $\mathcal{X}$ with a known TT structure as in Eq. 1, the task is to compute $\hat{\mathcal{X}} = \underset{\mathcal{A} \text{ as Eq. 1}}{\arg\min} \|\mathcal{X} - \mathcal{A}\|_F^2$. For the experiments, we first choose tensor modes and TT-rank with $R_D = 1$, generate $\mathcal{X}$ as in the TT-NF initialization scheme from Sec. 4.1 with $\sigma = 1.0$, and perform tensor contraction. To simulate noisy observations $\mathcal{Y}$, we sample noise $\mathcal{Z}$ from independent zero-mean Normal or Laplace distributions with a chosen scale and compute $\mathcal{Y} = \mathcal{X} + \mathcal{Z}$.

TT-SVD (Oseledets, 2011) is a deterministic[1] algorithm; it does not take any hyperparameters and produces a TT decomposition in a single pass over data. TT-OI (Zhou et al., 2022) performs several passes. TT-cross has several stopping settings, as it runs an iterative maximum volume algorithm (Goreinov & Tyrtyshnikov, 2001) on each iteration of the main algorithm. We take the defaults provided by the `tntorch` package (Usvyatsov et al., 2022).

On the side of the non-convex optimization family, we minimize a loss function between samples from the observation and TT-NF. We choose a loss function best suited for the type of observation noise: L1 for Laplace and L2 for Normal. Contraction denotes the usage of all possible elements in the loss, whereas Sampling considers only mini-batches of samples.

Fig. 4 demonstrates the performance of all methods on the problem of size $2^{20}$ under varying noise types, scale, and TT-NF rank. Optimization lasts 1000 steps, with a batch size of 4096 elements, Adam optimizer (Kingma & Ba, 2015) with default settings, learning rate warming up during the first 5% steps to $3e$-$2$ and decaying exponentially to $3e$-$4$. We ran all experiments 10 times and reported plots with 1-std error bars. All runs are completed on a single GPU.

TT-SVD provides a reasonable baseline, which we use as initialization for TT-NF. TT-OI works better than TT-SVD on a subset of Normal noise levels and ranks. TT-cross works only in a noise-free setting. Contraction, which can be considered as full-batch gradient descent, does not improve much upon TT-SVD. We conjecture that it gets stuck in saddle points, a recurring argument seen

---

[1]To the extent we can call SVD deterministic.

Table 2: Comparison of QTT-NF with NeRF by Mildenhall et al. (2020) and TensoRF by Chen et al. (2022). In the latter, we disable grid masking and bounding box fitting for even comparison with the other methods. We consider two techniques for obtaining view-dependent color: spherical harmonics (SH) with 28 channels and MLP (NN). For QTT-NF, we use a grid size of $256^3$. QTT-NF reaches performance competitive with the prior art. See Sec. 5.2 for more details.

| Metric | Method (shading) | Chair | Drums | Ficus | Hotdog | Lego | Materials | Mic | Ship | Avg. |
|---|---|---|---|---|---|---|---|---|---|---|
| PSNR ↑ | NeRF | 33.00 | 25.01 | 30.13 | 36.18 | 32.54 | 29.62 | 32.91 | 28.65 | 31.01 |
| | TensoRF (-mask) | 32.19 | 25.01 | 30.81 | 35.28 | 33.54 | 28.81 | 31.72 | 28.90 | 30.78 |
| | QTT-NF (SH) | 32.09 | 24.96 | 30.89 | 35.49 | 32.48 | 28.22 | 31.50 | 27.55 | 30.40 |
| | QTT-NF (NN) | 32.87 | 25.30 | 31.85 | 35.97 | 33.00 | 28.67 | 33.07 | 27.97 | 31.09 |
| SSIM ↑ | NeRF | 0.967 | 0.925 | 0.964 | 0.974 | 0.961 | 0.949 | 0.980 | 0.856 | 0.947 |
| | TensoRF (-mask) | 0.960 | 0.922 | 0.968 | 0.971 | 0.969 | 0.932 | 0.975 | 0.868 | 0.946 |
| | QTT-NF (SH) | 0.955 | 0.918 | 0.967 | 0.971 | 0.957 | 0.929 | 0.971 | 0.840 | 0.939 |
| | QTT-NF (NN) | 0.965 | 0.923 | 0.971 | 0.972 | 0.962 | 0.934 | 0.979 | 0.850 | 0.945 |
| LPIPS ↓ | NeRF | 0.046 | 0.091 | 0.044 | 0.121 | 0.050 | 0.063 | 0.028 | 0.206 | 0.081 |
| | TensoRF (-mask) | 0.053 | 0.094 | 0.037 | 0.055 | 0.040 | 0.075 | 0.029 | 0.170 | 0.069 |
| | QTT-NF (SH) | 0.062 | 0.095 | 0.039 | 0.060 | 0.066 | 0.093 | 0.039 | 0.203 | 0.082 |
| | QTT-NF (NN) | 0.050 | 0.094 | 0.037 | 0.055 | 0.052 | 0.089 | 0.032 | 0.201 | 0.077 |

in discussions of the advantages of SGD over GD in the deep learning literature. As can be seen, Sampling methods outperform all others in settings with the presence of noise.

## 5.2 NEURAL RADIANCE FIELDS

Following the fast-pacing domain, we test a QTT-NF variant of our representation (Fig. 2) in the neural radiance fields (NeRF) setting. The task of learning voxelized radiance fields from data assumes access to views of a single scene with known camera poses in some frame of reference. The objective is to regress features of the voxel grid such that when passed through differentiable shading and ray marching algorithms, the projected images would correspond to the ground-truth data, and views from the held-out set would exhibit high PSNR. Such a data-driven problem formulation does not permit the usage of existing SVD-based solutions and thus can only be solved through learning.

We start with the setting of Mildenhall et al. (2020) (recapped in Sec. A.2 of the Appendix) and replace the MLP converting coordinates and viewing direction into color and density with QTT-NF. Similar to Yu et al. (2022), we choose a voxel grid resolution of $256^3$ and 28 channels to store 9 spherical harmonics (SH) coefficients per RGB channel and one voxel density value.

Next, we compare QTT-NF with a recent work TensoRF (Chen et al., 2022), which likewise uses tensor decompositions, albeit triplanar ones. This work achieves remarkable reconstruction quality by using several state-of-the-art techniques on top of the proposed decomposition. We intentionally avoid such recipes that lead to better performance, voxel pruning, occupancy masks, extra losses (total variation), ray filtering, and progressive training schemes. One technique we borrow from that pool is the usage of a tiny MLP in place of SH for transforming voxel features into view-dependent color (NN), which retains the natural interpretability of the learned voxel grid features.

The standard testbed for neural rendering is the Blender dataset (Mildenhall et al., 2020)[2]. It consists of 8 synthetic 3D scenes, each with a hundred posed images of resolution $800 \times 800$. The scenes vary by complexity and include water and glossy surfaces, thus making it a challenging benchmark.

We train neural rendering experiments for 80K steps with an LR schedule similar to the one from Sec. 5.1, but with base LR $3e\text{-}3$, $R_{\max} = 256$, 4096 rays per batch, and 512 uniform samples per ray. The code of all experiments is implemented in PyTorch (Paszke et al., 2019). The observed variance of image quality metrics over five runs is not large: $\text{std}(\text{PSNR}) \sim 4e\text{-}2$, $\text{std}(\text{SSIM}) \sim 4e\text{-}4$, $\text{std}(\text{LPIPS}) \sim 5e\text{-}4$. Results from Tab. 2 attest that QTT-NF is capable of reaching performance competitive with the prior art.

---

[2]Distributed under different Creative Commons licenses. Per scene: "chair", "ficus", "hotdog", "materials", "mic": CC-0; "drums": CC-BY; "lego": CC-BY-NC; "ship": CC-BY-SA.

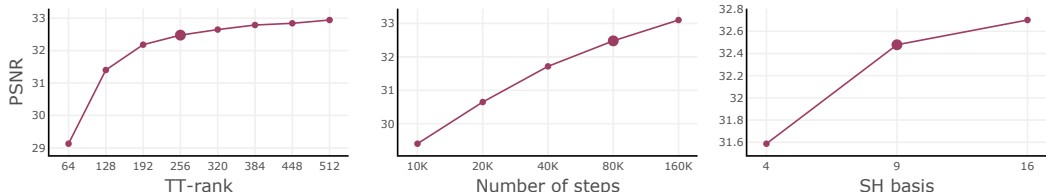

Figure 5: Sweeps of TT-rank, number of training steps, and spherical harmonics components per channel when training QTT-NF on the "Lego" scene. The remaining parameters are fixed to the baseline values (represented with a large dot in plots); see Sec. 5.2. Higher is better.

### 5.2.1 ABLATION STUDY

The ablation study of TT-rank, number of training steps, and SH basis in Fig. 5 shows that the results can be further improved, as most hyperparameters are not saturated. Additional studies of sampling schemes and pretraining with TT-SVD of a full voxel grid can be found in Sec. B of the Appendix.

The reconstruction quality of QTT-NF depends primarily on the TT-rank and scene complexity. This is contrary to TensoRF (Chen et al., 2022), which employs triplanar decompositions (CP, VM), and thus inevitably introduces a preference for axis-aligned content.

To demonstrate that, we introduce rotation of the scene around the Z-axis into the ground-truth poses of two scenes: "Lego" and "Hotdog". The former scene contains many axis-aligned primitives, which favorably utilize triplanar decomposition rank. We ensured that no rotation angle resulted in content out of voxel grid bounds. As can be seen from Fig. 6 (top), rotations of the same axis-aligned scene by 15, 30, and 45 degrees result in significant performance drops with triplanar decomposition and all ranks, whereas the performance of QTT-NF drops more gracefully. An ordinary scene from Fig. 6 (bottom) demonstrates the vanishing of this effect due to the absence of axis alignment in any of the rotations.

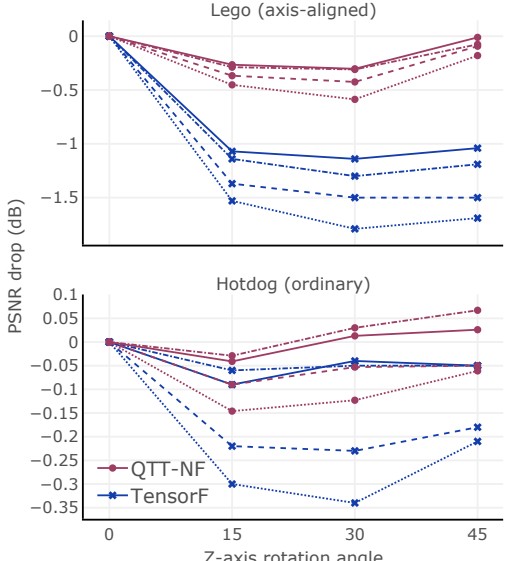

Figure 6: Sensitivity of QTT-NF and TensoRF (Chen et al., 2022) to context-to-axes alignment. Scenes: top – "Lego", bottom – "Hotdog". TensoRF exhibits a performance drop with all ranks when rotating axis-aligned scenes. Pattern: dot – 25%, dash – 50%, dashdot – 75%, solid – 100% of the rank. Higher is better.

The average performance of QTT-NF (with 2.16M parameters) is slightly higher than TensoRF (with 3.17M parameters). This attests to better parameter-efficiency of QTT-NF for scene representation.

## 6 CONCLUSION

We presented TT-NF, a novel low-rank representation for neural fields that can be learned directly via backpropagation through samples and optimization. Our representation avoids instantiating the full uncompressed tensor but instead learns it in a compressed form by optimizing a non-convex objective defined by the target task. We applied TT-NF to a synthetic tensor denoising task, where we outperformed standard SVD-based approaches and the real-world novel view synthesis problem. For the latter, we proposed QTT-NF, a modification of TT-NF that handles hierarchical spaces, such as 3D voxel grids. Last but not least, we proposed efficient sampling algorithms for our neural fields and showed that these algorithms increase efficiency both w.r.t. speed and memory, making (Q)TT-NF friendly for applications such as rendering. We further direct interested readers to Sec. C of the Appendix for extra discussion.

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

# A    METHOD

## A.1    RANKS AND PARAMETERIZATIONS

**TT-rank Selection**    We clarify the procedure of choosing a TT-rank, mentioned in Sec. 4.1. To parameterize TT-NF, we need to know the dimensions (modes) of the tensor $M_i, i = \overline{1, D}$, the payload dimension $R_D$, and a scalar hyperparameter $r$, defining the degree of compression. A TT-rank resulting from a TT-SVD (Oseledets, 2011) algorithm is bounded as in Holtz et al. (2012, Eq. 20), with the only difference in added $R_D$ term to account for the discussed block structure:

$$1 \leq R_k \leq R_k^{\max} \equiv \min \left( \prod_{i=1}^{k} M_i, \ R_D \prod_{i=k+1}^{D} M_i \right), \ \ k = \overline{1, D\text{-}1}. \tag{3}$$

This means that the maximal TT-rank forms a "pyramid" of integers, where each position $i$ is assigned a minimum of products of modes to the left and right of $i$. We denote $R_{\max}$ as the maximum value of TT-rank and $i_{\max}$ as the position of the maximum value (the "peak"). As shown in Fig. 3, 7, and 8, $R_{\max}$ can go as high as 1024 in $2^{20}$ and 32768 in $2^{30}$ sizes. To achieve compression with our parameterizations, we clamp the "pyramid" at a chosen value of rank $r \in [1, R_{\max}]$, thus making the TT-rank "trapezoid". This range corresponds to the X-axes of the aforementioned figures.

**Full TT-NF Parameterization**    Full parameterization assumes the allocation of learned parameters to each element of TT-cores $\mathcal{C}^{(i)}, i = \overline{1, D}$. Such parameterization is compatible with **v1**, **v2** sampling schemes, as well as tensor contraction and subsampling.

**Reduced TT-NF Parameterization**    A TT manifold of a fixed TT-rank has a certain number of Degrees of Freedom (DOF) (Holtz et al., 2012), which is smaller than the number of learned parameters of TT-NF with **v2** sampling. Practical mappings from DoF to learned parameters exist but require complex transformations (Obukhov et al., 2021). The latter work provides insights into a simpler form of reduced parameterization, which becomes possible once the rank becomes large enough that some TT-cores have square "matricizations", meaning that either $R_{i-1} M_i = R_i$ or $R_{i-1} = M_i R_i$. Such cores can effectively be replaced with fixed identity matrices (reshaped into the original core shapes), leading to parameter count reduction. Additionally, knowledge of such reduced parameterization suggests a modification to the **v2** sampling algorithm, aliased **v3**.

Given $r$, we denote $1 \leq p \leq q \leq D$ indices of the first and last clamped values in TT-rank. It is easy to verify that TT-cores $\mathcal{C}^{(i)}$ to the left of $p$ and to the right of $q$ should have square matricizations. Indeed, a TT-core at position $i < p$ has a shape $R_{i-1} \times M_i \times R_i = R_{i-1}^{\max} \times M_i \times R_i^{\max} = \left( \prod_{j=1}^{i-1} M_j \right) \times M_i \times \left( \prod_{j=1}^{i} M_j \right)$, thus its left matricization (merging the first two dimensions, denoted `leftmat`) is square.

As shown in Obukhov et al. (2021), such cores can effectively be replaced with fixed identity matrices (reshaped into the original core shapes) without any loss of representation power of such TT-NF. The sparsity of identity matrices suggests the possibility of skipping steps of the **v2** sampling algorithm and changing several matrix multiplication operations (`Linear` layers) to a single indexing operation.

Considering the left side of such a TT-NF again, if $\mathcal{C}_{1,:,:}^{(1)} = I_{R_1}$, then on the first step of Alg. 3 $v = e_{i_1}$, where $e_j$ denotes an ort with 1 in $j$-th position. By induction, if $v = e_{i_{\text{left}}}$ and $\texttt{leftmat}(\mathcal{C}^{(k)}) = I_{R_k}$, then $\mathcal{C}_{:,i_k,:}^{(k)} v = e_{i_{\text{left}} M_k + i_k}$. Such index propagation rule allows us to skip `BIMVP` computation for all such cores and index $v$ directly along the first rank mode of the first parameterized TT-core at position $p$. The index propagation rule for the right side of TT-NF can be derived similarly. The full algorithm for **v3** sampling can be found in Alg. 4.

Notably, in the extreme case of decomposition rank $r = R_{\max}$, only a single TT-core contains learned parameters, effectively of an uncompressed tensor. In this case, sampling with **v3** becomes the indexing operation along all three modes of the TT-core, leading to zero FLOPs. This corner case explains the drop of FLOPs line around the far end of the rank axis. The remaining saw-like drops occur when $r$ increases above intermediate values, making one or two more TT-cores become reshaped identity matrices. While the absolute gain in FLOPs is not substantial when rank $r$ is not

**Algorithm 3** Recap of **v2** sampling explained in Sec. 4.2 (for comparison with **v3** on the right).

**Require:**
  $D$ - number of tensor dimensions,
  $B$ - number of samples,
  $(1, R_1, ..., R_D)$ - TT-rank,
  $(M_1, ..., M_D)$ - TT-modes,
  $(\mathcal{C}^{(1)}, ..., \mathcal{C}^{(D)})$ - TT-cores representing $\mathcal{A}$,
  $\left( \left( i_1^{(1)} ... i_D^{(1)} \right), ..., \left( i_1^{(B)} ... i_D^{(B)} \right) \right)$ - indices.

**Ensure:**
  $v = \left( \mathcal{A}_{i_1^{(1)}, ..., i_D^{(1)}}, ..., \mathcal{A}_{i_1^{(B)}, ..., i_D^{(B)}} \right)$ - samples
  without computing the whole $\mathcal{A}$.

1: $\pi \leftarrow (1, ..., B)$     ▷ forward permutation
2: $\sigma \leftarrow (1, ..., B)$     ▷ inverse permutation
3: $v \leftarrow \mathcal{C}_{1, i_1, :}^{(1)}$        ▷ $B \times R_1$
4: **for** $k \leftarrow 2$ to $D$ **do**
5:    $i_k \leftarrow \pi(i_k)$     ▷ align mode indices
6:    $v, \pi_k \leftarrow \mathtt{BIMVP}(\mathcal{C}^{(k)}, i_k, v)$    ▷ $B \times R_k$
7:    $\sigma_k \leftarrow \pi_k^{-1}$     ▷ invert $k^{\text{th}}$ permutation
8:    $\pi \leftarrow \pi_k(\pi)$     ▷ update forward perm.
9:    $\sigma \leftarrow \sigma(\sigma_k)$     ▷ update inverse perm.
10: **end for**
11: $v \leftarrow \sigma(v)$     ▷ recover samples order
12: **return** $v$       ▷ $B \times R_D$

Intentionally left blank

**Algorithm 4** Memory-efficient Sampling from a Reduced Parameterization of TT-NF (**v3**). Auto-differentiation paths are highlighted in blue. Refer to Sec. A.1 for more details.

**Require:**
  $D$ - number of tensor dimensions,
  $B$ - number of samples,
  $1 \le p \le q \le D$ - parameterized cores range,
  $(1, R_1, ..., R_D)$ - TT-rank,
  $(M_1, ..., M_D)$ - TT-modes,
  $(\mathcal{C}^{(p)}, ..., \mathcal{C}^{(q)})$ - TT-cores representing $\mathcal{A}$,
  $\left( \left( i_1^{(1)} ... i_D^{(1)} \right), ..., \left( i_1^{(B)} ... i_D^{(B)} \right) \right)$ - indices.

**Ensure:**
  $v = \left( \mathcal{A}_{i_1^{(1)}, ..., i_D^{(1)}}, ..., \mathcal{A}_{i_1^{(B)}, ..., i_D^{(B)}} \right)$ - samples
  without computing the whole $\mathcal{A}$.

1: $i_{\text{left}} \leftarrow (0, ..., 0)$     ▷ $B$ left indices
2: $i_{\text{right}} \leftarrow (0, ..., 0)$     ▷ $B$ right indices
3: **for** $k \leftarrow 1$ to $p - 1$ **do**
4:    ▷ Propagate left indices
5:    $i_{\text{left}} \leftarrow i_{\text{left}} * M_k + (i_k - 1)$
6: **end for**
7: **for** $k \leftarrow q + 1$ to $D$ **do**
8:    ▷ Propagate right indices
9:    $i_{\text{right}} \leftarrow i_{\text{right}} + (i_k - 1) * R_k$
10: **end for**
11: $i_{\text{left}} \leftarrow i_{\text{left}} + 1$     ▷ make start from 1
12: $i_{\text{right}} \leftarrow i_{\text{right}} + 1$     ▷ make start from 1
13: **if** $p = q$ **then**
14:    ▷ Directly index $v$ in $\mathcal{C}^{(p)}$
15:    $v \leftarrow \mathcal{C}_{i_{\text{left}}, i_p, i_{\text{right}}:i_{\text{right}}+R_D}^{(p)}$
16: **else**
17:    $\pi \leftarrow (1, ..., B)$    ▷ forward permutation
18:    $\sigma \leftarrow (1, ..., B)$    ▷ inverse permutation
19:    $v \leftarrow \mathcal{C}_{i_{\text{left}}, i_p, :}^{(p)}$       ▷ $B \times R_p$
20:    **for** $k \leftarrow p + 1$ to $q$ **do**
21:      $i_k \leftarrow \pi(i_k)$     ▷ align mode indices
22:      $v, \pi_k \leftarrow \mathtt{BIMVP}(\mathcal{C}^{(k)}, i_k, v)$   ▷ $B \times R_k$
23:      $\sigma_k \leftarrow \pi_k^{-1}$     ▷ invert $k^{\text{th}}$ permutation
24:      $\pi \leftarrow \pi_k(\pi)$     ▷ update forward perm.
25:      $\sigma \leftarrow \sigma(\sigma_k)$     ▷ update inverse perm.
26:    **end for**
27:    $v \leftarrow \sigma(v)$     ▷ recover order, $B \times R_q$
28:    $v \leftarrow v_{:, i_{\text{right}}:i_{\text{right}}+R_D}$     ▷ $B \times R_D$
29: **end if**
30: **return** $v$      ▷ $B \times R_D$

large, we observe a noticeable speed-up of $2$-$4\times$ with **v3**, compared to training with **v2**, due to fewer sequential invocations of `Linear` layer functions.

The reduced parameterization is compatible with **v1**, **v2**, **v3** sampling schemes, as well as tensor contraction and subsampling.

**Conversion from Full to Reduced Parameterization**    Given a Full parameterization, conversion to a Reduced one can be performed in a single pass over TT-cores. Indeed, starting from the first TT-core

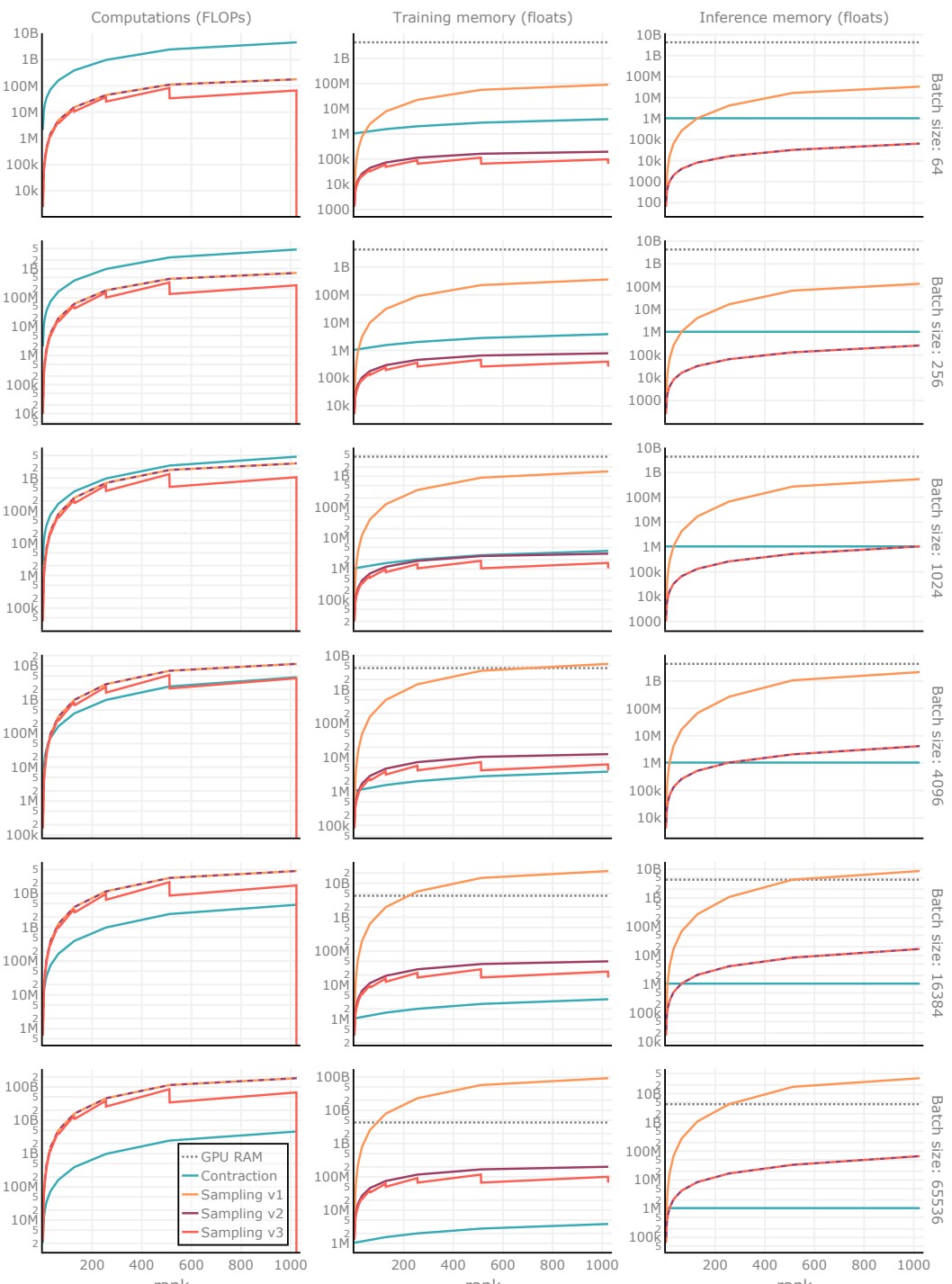

Figure 7: Space-time complexity of sampling from TT-NF of size $2^{20}$ with various methods, batch sizes, and ranks. We compare three sampling schemes discussed in Sec. 4, A.1, as well as the traditional tensor contraction scheme. As seen from the plots, the optimal choice of sampling scheme depends on the rank, problem, and batch sizes. Lower is better. Best viewed in color.

and repeating until reaching position $p$, we can compute the left matricization of the current TT-core and absorb (through matrix multiplication) this square matrix into the right-hand-side TT-core. The

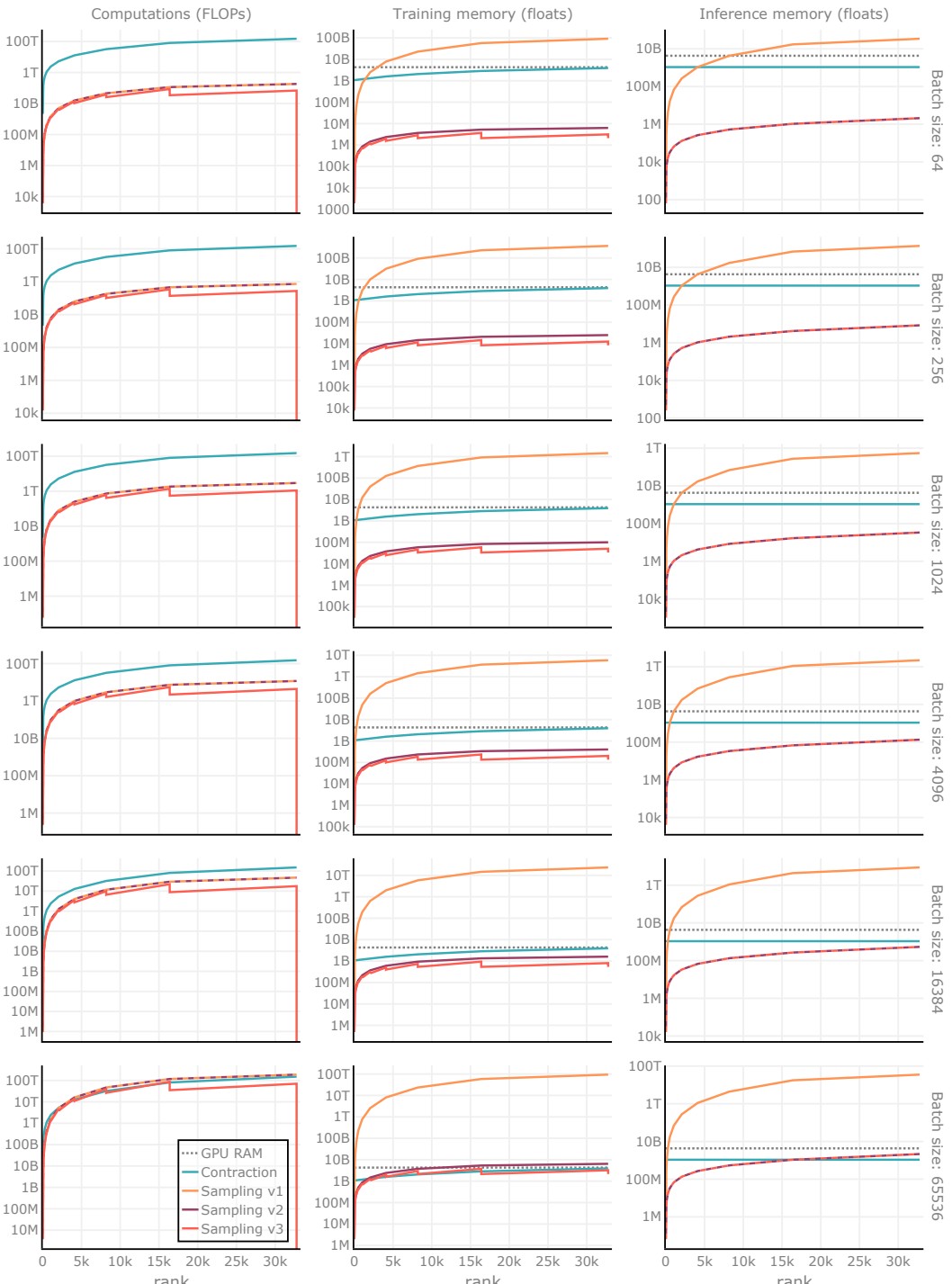

Figure 8: Space-time complexity of sampling from TT-NF of size $2^{30}$ with various methods, batch sizes, and ranks. We compare three sampling schemes discussed in Sec. 4, A.1, as well as the traditional tensor contraction scheme. As seen from the plots, the optimal choice of sampling scheme depends on the rank, problem, and batch sizes. Lower is better. Best viewed in color.

current core is then replaced with an identity matrix and reshaped into the original shape of the core. A similar process starting from the last TT-core until reaching position $q$ completes the conversion.

### A.2 NEURAL RENDERING

The core of neural rendering is comprised of two main components: (1) a neural field and (2) differentiable ray marching.

**Neural Field** In the paper by Mildenhall et al. (2020), the neural field is represented with a neural network mapping $f_\theta : (x, d) \to (c, \sigma)$, where $x$ is a position in some scene-centric frame of reference, $d$ is a viewing direction pointing at a camera, $c$ is a view-dependent color (e.g., 8-bit RGB channels), and $\sigma$ is a view-independent volume density at $x$.

Other works (Yu et al., 2022) popularized the usage of spherical harmonics to represent view-dependent color variations. Effectively, the viewing direction $d$ is removed from the parameterization; instead, color descriptors of sufficient size are regressed at each location along with density $\sigma$. Such color descriptors can be coefficients in a pre-defined spherical harmonics basis, effectively parameterizing color as a function on a sphere. Alternatively, the transformation of the learned descriptor and viewing direction can be learned with a dedicated "shading" MLP, as done in Chen et al. (2022). Empirically, degree two harmonics appear sufficient to represent view-dependent color variations with high fidelity, which translates into a per-voxel payload size of 28: one value to represent density and three groups of nine coefficients per channel.

**Ray Marching** Image formation with radiance fields is usually accomplished through a differentiable ray marching procedure, which is a simplistic version of ray-tracing. Given camera calibration, a ray is cast through each pixel of the output image to accumulate color at intersections with the neural field. Samples from the neural field are taken along each ray guided by prior knowledge of how the ray passes through the field and where the scene roughly is. In NeRF (Mildenhall et al., 2020), 64 samples are sampled uniformly to probe the field between hardcoded near and far planes, and then 128 more samples are obtained to increase sampling density at places with content. Voxel grid neural fields (Yu et al., 2022) alter the sampling procedure: rays are filtered based on whether they intersect with the voxel grid, and points inside the grid are sampled uniformly between near and far intersection points. A floating-point coordinate sampling of voxel grids is resolved through trilinear interpolation of payload vectors. We follow the same procedure in our experiments with QTT-NF.

Samples $(c_i, \sigma_i)$ at positions $r_i$ of a ray $r$ are accumulated using the following equations:

$$C(r) = \sum_{i=1}^{N} T_i \left(1 - \exp(-\sigma_i \delta_i)\right) c_i, \text{ where } T_i = \exp\left(-\sum_{j=1}^{i-1} \sigma_j \delta_j\right).$$

Here $\delta_i = \|r_i - r_{i-1}\|_2$, $T_i$ represents optical transmittance of the ray at position $r_i$, and the last term denotes contribution of the position to the accumulated color.

## B EXPERIMENTS

### B.1 EFFECT OF REDUCED PARAMETERIZATION

We compare our reduced (**v3**) parameterization with the full one (**v2**) in the default QTT-NF setting with spherical harmonics and report results in Tab. 3 (top). As can be seen, the average PSNR does not change much depending on parameterization. This is contrary to the tensor denoising case described in Sec. 5.1, which we can attribute to a different (longer) training regime.

### B.2 INITIALIZATION OF QTT-NF USING TT-SVD

This set of experiments demonstrates the positive effect of QTT-NF initialization from an uncompressed voxel grid. We start by training such uncompressed representations with the same grid configuration ($256^3 \times 28$ parameters). The training protocol is the same as for QTT-NF training, except for the usage of larger yet different learning rates for the parameters of spherical harmonics ($1e\text{-}1$) and density ($1e1$). The results can be seen in Tab. 3 (Uncompressed); many scenes receive lower scores than training compressed representations from random initialization due to the lack

Table 3: Top: per-scene comparison of QTT-NF trained with **v2** and **v3** from random initialization (Sec. B.1); Bottom: experiments with training uncompressed voxel grid and using it as initialization for QTT-NF through TT-SVD (Sec. B.2); Unlike the tensor denoising setting, in neural rendering, both **v2** and **v3** parameterizations produce almost identical results. Initialization from uncompressed volume improves performance.

| Metric | Method (flavor) | Chair | Drums | Ficus | Hotdog | Lego | Materials | Mic | Ship | Avg. |
|---|---|---|---|---|---|---|---|---|---|---|
| PSNR ↑ | QTT-NF (**v2**) | 31.88 | 24.95 | 30.72 | 35.30 | 32.17 | 28.23 | 31.58 | 27.84 | 30.33 |
| | QTT-NF (**v3**) | 32.09 | 24.96 | 30.89 | 35.49 | 32.48 | 28.22 | 31.50 | 27.55 | 30.40 |
| | Uncompressed | 32.54 | 25.10 | 31.30 | 33.53 | 31.91 | 27.63 | 32.38 | 22.76 | 29.64 |
| | QTT-NF (init unc.) | 32.26 | 25.09 | 30.93 | 35.53 | 32.76 | 28.58 | 32.18 | 27.93 | 30.66 |
| SSIM ↑ | QTT-NF (**v2**) | 0.953 | 0.918 | 0.967 | 0.970 | 0.956 | 0.929 | 0.971 | 0.843 | 0.938 |
| | QTT-NF (**v3**) | 0.955 | 0.918 | 0.967 | 0.971 | 0.957 | 0.929 | 0.971 | 0.840 | 0.939 |
| | Uncompressed | 0.967 | 0.925 | 0.970 | 0.960 | 0.959 | 0.926 | 0.979 | 0.785 | 0.934 |
| | QTT-NF (init unc.) | 0.958 | 0.921 | 0.968 | 0.972 | 0.961 | 0.934 | 0.976 | 0.845 | 0.942 |
| LPIPS ↓ | QTT-NF (**v2**) | 0.065 | 0.098 | 0.040 | 0.063 | 0.067 | 0.093 | 0.041 | 0.209 | 0.085 |
| | QTT-NF (**v3**) | 0.062 | 0.095 | 0.039 | 0.060 | 0.066 | 0.093 | 0.039 | 0.203 | 0.082 |
| | Uncompressed | 0.042 | 0.077 | 0.038 | 0.071 | 0.047 | 0.088 | 0.025 | 0.223 | 0.076 |
| | QTT-NF (init unc.) | 0.059 | 0.093 | 0.037 | 0.058 | 0.058 | 0.087 | 0.031 | 0.202 | 0.078 |

of L1 or TV regularization. Further, we apply dimensions factorization of the uncompressed voxel grid to match the decomposition scheme from Fig. 2 and perform TT-SVD with rounding to the TT-rank matching our QTT-NF. The resulting decomposition is loaded into QTT-NF instead of random initialization per Eq. 2, and the same training protocol (only with a lower learning rate of $1e\text{-}3$) is executed. The results of such fine-tuning are presented in Tab. 3 (QTT-NF init unc.). Evidently, TT-SVD provides a good initialization for QTT-NF, best in L2 distance between the approximation and the uncompressed volume, yet suboptimal in terms of the downstream task performance. This suboptimality is corrected through fine-tuning, and the resulting performance exceeds that of training from random initialization.

# C  DISCUSSION

**QTT – Quantized or Quantics?**   The concept of representing data as low-rank tensor decompositions with the logarithmic number of modes and particular mode grouping pattern was first discussed in Oseledets (2009); Khoromskij (2009), the latter work coining the term "Quantics Tensor Train", used up until present days, e.g., Oseledets & Tyrtyshnikov (2011); Soley et al. (2021). Several other works (Dolgov et al., 2012; Kazeev et al., 2017; Poirier et al., 2020) call the same concept "Quantized Tensor Train", thus making these terms interchangeable. The term means factorizing (or quantizing) space into repeating small factors ("quants"). The term "quantization" in machine learning often refers to the reduction of the dynamic range of values of learned parameters in a parametric model. This exact terminology clash seen in QTTNet (Lee et al., 2021) suggests that the usage of the "Quantics" variant might lead to less ambiguity in the considered context.

**Supported Parallelism**   All discussed sampling schemes support data parallelism: TT-NF parameters can be replicated across multiple computing devices, and a batch of samples is split among them. This is contrary to tensor contraction, where the main constraint is to be able to fit the entire uncompressed tensor into memory. On the other hand, optimized contraction schemes (Smith & Gray, 2018; Rogozhnikov, 2022) followed by subsampling are expected to outperform all sampling schemes starting with some sufficiently large batch size.

**Low-level Optimizations**   The proposed sampling schemes (Alg. 1,4) are designed to perform well at training time in all off-the-shelf deep learning frameworks with automatic differentiation support while keeping memory pressure low. Permutations enable the use of the standard `Linear` layer, which is one of the first functions undergoing heavy optimization on any new-generation computational device, along with memory operations such as permutations themselves, BLAS (Blackford et al., 2002), convolutions (LeCun et al., 1989), normalization layers (Ioffe & Szegedy, 2015), and activation

functions (Nair & Hinton, 2010). It is possible, however, to get rid of permutations through a custom implementation of the BIMVP function (Alg. 2), thus obtaining $2\times$ reduction in space requirements. Shi et al. (2016) propose a StridedBatchedGemm BLAS primitive, which would need to accept a batch of offsets instead of scalar strides. CUTLASS (Kerr et al., 2017) provides more promising building blocks for implementing both forward and backward passes. Deploying TT-NF sampling for inference in production and edge devices can be feasible, provided the presence of AI accelerators optimized for neural network inference. An overview of the current state of mobile computing is given in Ignatov et al. (2019).

**Limitations**   In short training protocols such as experiments with tensor denoising (Sec. 5.1), **v3** sampling consistently does not reach the performance of **v2**, despite effectively learning the same representation. We hypothesize that this effect may have an explanation in the deep linear networks literature. This effect is not pronounced in the NeRF setting with its longer training protocol, as seen in Tab. 3. The reduced complexity of **v3** can nevertheless help with saving computational resources during inference sampling via conversion from **v2**, as explained in the last paragraph of Sec. A.1.

**Future Research Directions**   The underlying Tensor Train representation of TT-NF permits many interesting usage scenarios yet to be explored. For example, the TT-format can be used to quickly compute marginals over selected modes $\{M_i\}$, which could be used to compute level-of-detail (LOD) samples with QTT-NF. Representation capacity can be controlled dynamically through rank rounding (with TT-SVD) or expansion (by padding TT-cores). TT-NF may be found useful in a streaming setting, where the neural field is continuously updated.

