# OpenReview forum: "TT-NF: Tensor Train Neural Fields"
_ICLR.cc/2023/Conference — Submitted to ICLR 2023_

### Official Review · Reviewer_wzYM · 2022-10-17

**Confidence:** 4
**Correctness:** 4
**Technical Novelty And Significance:** 2
**Empirical Novelty And Significance:** 3
**Recommendation:** 5

**Clarity, Quality, Novelty And Reproducibility:**

**Clarity and Quality:**

1. Following Eq. (2), there should be proofs to say why $\hat{\sigma}$ guarantees the variance of the full tensor.
2. In the experiment of tensor denoising, the proposed method outperforms TT-SVD and others, but there is no discussion on where the advantage comes from. One reason, I guess, is that the TT-SVD assumes the semi-orthogonality of the TT-cores, but the proposed method does not. It makes the TT-SVD is representationally weaker than the proposed one due to the additional constraints. Therefore, I doubt if the experiment of tensor denoising in the paper can support the central claim properly.
3. For the second experiment of neural radiance fields, Tab 2 shows that the proposed method is only comparable with the existing methods. I am thus confused about the main advantages of the new methods then. If there is another advantage, such as computational efficiency or interpretability, it should be clearly mentioned in this part.

=========================
*Update*:
1. " One reason, I guess, is that the TT-SVD assumes the semi-orthogonality of the TT-cores, but the proposed method does not. It makes the TT-SVD is representationally weaker than the proposed one due to the additional constraints."

-- I am convinced that the additional semi-orthogonality will not affect the representational power of the TT model. In other words, it is fair to say the TT-SVD, TT-OI, Contraction and the proposed v2 and v3 in Fig. 4 have similar representational power, and the main differences among these methods are optimization schemes. Based on this point, it is not clearly explained why v2/3 outperforms other methods a lot in Fig. 4. Although the authors mentioned that Contraction might get stuck in saddle points, it cannot convince me due to the lack of more detailed investigation.

**Novelty:**

Please check the “Strength and Weakness” section.

**Reproducibility**

The experimental settings were introduced clearly.

**Strength And Weaknesses:**

++Strength

The part of “sampling from TT-NF” introduced a practically valuable trick to reduce the storage complexity when calculating the (partial) contraction of tensor networks.

— Weakness

1. Although a lot of literature was overviewed in the “Related work” part, for each highlighted aspect, the overview is not sufficient, especially for the parts “tensor decomposition” and “neural network compression with tensors.”
2. The novelty is relatively weak. In Section 4 — “Method” — the paper introduced an initialization of TT-NF in Sec 4.1 and the sampling trick from TT-NF in Sec 4.1. For the former, the variance-preserving core initilization has been discussed in (Pan Y et al., 2022); for the latter, it is practically useful (mentioned in Strength), but the trick seems trivial in my own opinion.

Pan, Yu, et al. "A Unified Weight Initialization Paradigm for Tensorial Convolutional Neural Networks." International Conference on Machine Learning. PMLR, 2022.

**Summary Of The Paper:**

In this paper, the authors proposed a new method, named TT-NF, for learning neural fields in terms of tensor train networks. Technically, a storage-efficient algorithm for sampling from TT-NF was introduced: the storage complexity is reduced from O(BDR^2) to O(BDR), where R denotes the maximum bond dimension (TT-ranks). Empirically, the new method was applied to the task of neural radiance fields. The numerical results showed that the proposed methods achieve competitive performance with SOTAs.

**Summary Of The Review:**

The trick on “sampling from TT-NF” is practically useful for other tensor researchers to implement their codes on computers more efficiently. The drawbacks of this paper are also obvious: (relatively) weak novelty and unconvincing experimental results. My recommendation is thus on the boundary.

---

> ### Author Response · Authors · 2022-11-16
> **Conclusion**
>
> We thank the reviewer for their time and effort in reading our submission and writing the review. Hopefully, our response addresses the main concerns stated in the review. We will implement the discussed improvements in the next revision of the paper. We urge the reviewer to evaluate our feedback and consider increasing their score.

---

> > ### Comment · Reviewer_wzYM · 2022-11-19
> > **Thanks for the responses.**
> >
> > I revised the reviews and increased the score on correctness (3->4) and empirical novelty (2->3). I still maintained the overall score since the quality of the paper was on the boundary but still relatively lower than some other papers I reviewed. I'm also okay if the paper is accepted with careful revision on the clarity.

---

> ### Author Response · Authors · 2022-11-16
> **Advantages of (Q)TT-NF over other Neural Radiance parameterizations**
>
> While we mention that neural radiance parameterization is just an example application for us and that we focus on the advantages and limitations of the proposed parameterizations, we also draw the reviewer’s attention to the fact that our parameterization:
> - is more parameter-efficient than other types of differentiable sparsity,
> - supports algebraic operations in the low-rank format,
> - can be initialized and fine-tuned from the uncompressed representation, and
> - allows upscaling to intractable volumes, which are otherwise impossible to store in the computer memory.
>
> Logarithmic scalability in the linear dimensions of the represented lattice is what makes this representation unique.

---

> ### Author Response · Authors · 2022-11-16
> **Discussion of advantages over TT-SVD**
>
> (Semi-)orthogonality of the TT-cores has little to do with the expressivity of the TT manifold. Holz et al. state and prove the exact dimensionality of the tensor train manifold of a fixed rank; our work provides a way to parameterize this manifold, and to convert to and from the TT-SVD initialization, v2-, and v3-compatible representations. See Section A.1 “Conversion from Full to Reduced Parameterization” for details.
>
> The real advantage of TT-NF over TT-SVD and TT-cross is that the non-convex optimization setting allows for noise in the observations by choosing a loss function, whereas previous algorithms are guaranteed to output the low-rank structure only without noise. Whether these guarantees hold in practice or not is explored empirically in our paper, particularly in Figure 4.

---

> > ### Comment · Reviewer_wzYM · 2022-11-19
> > **Sorry for my misunderstanding. I agree with this point.**
> >
> > Thanks for the explanation! Yes, you are right. (Semi-)orthogonality on TT cores will not decrease the expressive capability of the TT model. I will revise my reviews.

---

> ### Author Response · Authors · 2022-11-16
> **Proofs for Eq. (2)**
>
> We would be thrilled if our paper would prompt follow-up works that develop a formal theory of the initialization, taking into account the covariance of the parameters. Yet this is beyond the scope of our work. Recall that much of the recent progress in representation learning was only formally grounded ex-post.

---

> ### Author Response · Authors · 2022-11-16
> **Novelty**
>
> We respectfully disagree with the reviewer’s claim about weak novelty. To the best of our knowledge, our work is the first to show how to update the TT parameterization via gradient-based optimization efficiently in the stochastic setting. That is, we introduce a novel, learned representation and algorithms to work with it, which perfectly aligns with the scope of ICLR. If the reviewer is aware of any prior work that does this, we would be happy if they could point us to a concrete reference.

---

> > ### Comment · Reviewer_wzYM · 2022-11-19
> > **Regarding the novelty, it's on the boundary on my side.**
> >
> > "For the latter, it is practically useful (mentioned in Strength), but the trick seems trivial in my own opinion".
> >
> > --  I agree that the sampling trick introduced in sec 4.2 was practically useful. I also tend to believe it is new to utilize the batched matrix multiplication (BMM)  combing with the sampling and subtle permutation to reduce the computational cost for the TT decomposition. However, at a general level, using automatic differentiation for tensor network decomposition is quite standard (also mentioned by other reviewers) in the ML community. Consequently, various loss functions can be naturally chosen to deal with different types of noise.
> >
> > To sum up, I agree the trick of "sampling from TT-NF" is useful, particularly when someone wants to run large-scale tensor train decomposition on GPUs. But for other parts, the novelty of this paper is lower than the average of the ICLR papers I reviewed.
> >
> > I know the discussion of novelty is quite subjective since different people have different favorites. I believe the AC can balance the different ideas of authors, reviewers and others.

---

> ### Author Response · Authors · 2022-11-16
> **Initialization trick**
>
> Indeed, the initialization trick is not novel at all and is borrowed from the related work on (C)NN weight matrices initialization, such as He and Glorot. We thank the reviewer for the suggestion, and we will cite the suggested work in the overview of network compression. Parameter initialization is a small part compared to the overall scope: the proposed representation, efficient sampling algorithms, and insights into training with deep learning tools. Thus, we do not see why the initialization trick would detract the reviewer from the novelty of our work.

---

> > ### Comment · Reviewer_wzYM · 2022-11-19
> > **"why the initialization trick would detract the reviewer from the novelty of our work."**
> >
> > Thanks for the response. Let me explain more on this point. The initialization trick is not the (main) factor that detracts me from the novelty of your work, but it is still deserved to be mentioned in the review.
> >
> > "For the former, the variance-preserving core initialization has been discussed in (Pan Y et al., 2022); "
> >
> >   -- Like you are confused by the review, I was also confused when I first went through the paper. Why did the authors highlight this "quite small part" (described by authors) by a whole subsection, putting it at the beginning of Section 4, and with almost half a page?  These facts made me recognize that the authors wanted to highlight the initialization trick as one of the main contributions of the work. Now I understand it is just a "quite small part" as mentioned by the authors. Such confusion can be eliminated by improving the clarity of  sec 4, such as simply exchanging sec 4.1 and sec 4.2. Another clarity issue is that there is *no* explanation about how Eq. (2) is achieved, like any deduction or any citations, no matter Pan or He's work or the one mentioned by Ivan. Taken together, I thought this point needed to be mentioned in the review due to the confusing literal expression and the lack of explanation.

---

> ### Author Response · Authors · 2022-11-16
> **Missing references**
>
> We thank the reviewer for the suggestion. We will include the mentioned reference in our NN compression literature review. We also invite the reviewer to share other works that they think would make our review of related work complete.

---

> ### Public Comment · ~Ivan_Oseledets1 · 2022-11-18
> **TT-SVD representational power**
>
> It is well-known that TT decomposition is not unique, and orthogonality conditions do not reduce representation power of it (i.e. every TT representation can be written is the orthogonal form), so at least one of the reviewer statements is factually incorrect.
> The variance preserving trick has been introduced not in Pan, Yu, but in our paper with Valentin Khrulkov on tensorized embeddings >2 years ago.
>
>
> I also don't agree on the novelty: I think, the paper solves one of the important drawbacks in using tensor decompositions regarding samples.

---

> > ### Comment · Reviewer_wzYM · 2022-11-19
> > **Thanks for pointing out.**
> >
> > Dear Ivan,
> >
> > Many thanks for pointing out the mistakes in my review (also mentioned in the authors' response). You are right. I have revised the review and updated the score according to this point. As for the novelty, I reply to it following the authors' post.  Please find it if you are interested. In summary, I acknowledge the good contribution of this work on the practical side (for those who would like to implement large-scale TT decomposition on GPUs), but still concerned that the novelty of this work is relatively lower than the average.
> >
> > Best regards,

---

### Official Review · Reviewer_woGR · 2022-10-25

**Confidence:** 3
**Clarity, Quality, Novelty And Reproducibility:** Overall, the paper is clear, and shou…
**Correctness:** 2
**Technical Novelty And Significance:** 2
**Empirical Novelty And Significance:** 2
**Recommendation:** 3

**Strength And Weaknesses:**

 Strength:

- The idea of incorporating tensor train decomposition to neural fields is exciting, albeit it is a natural way when thinking of different tensor decompositions beyond the CP/VM ones proposed in Tensorf [Chen et al. 2022]

- The sampling algorithm does alleviate the computational issue with the TT formulation

Weakness:

- One major issue of using such TT form is that it involves many matrix multiplications instead of the element-wise multiplications used in CP/VM decomposition as in [Chen et al. 2022]. Therefore, the computational complexity of a single sampling point is O(Dr^3) compared with O(Dr) in CP, where r is the internal rank. Based on my understanding, the sampling method only aims for reducing the space complexity during the inference, and relaxes the matrix multiplication to vector-matrix multiplication. However, the vector-matrix multiplication is still **inefficient** compared with the point-wise multiplication in CP/VM as in [Chen et al. 2022]. Moreover, for back-propagation during training, I believe this would take more GPU memory compared with CP/VM.

- The experimental results on Synthetic NeRF dataset is not convincing. In particular, it apparently has inferior performance than TensoRF in both SSIM and LPIPS. Even for PSNR, I don't see a clear advantage since in the original paper of TensoRF the reported quantitative performance is much higher than that of reported here. Although it is said in the paper that QTT-NF intentionally avoid voxel pruning, total variation loss, etc, I don't think this is a reasonable argument since we can also adopt the same practice here to see if QTT-NF can achieve the similar results under the same setting. Moreover, I know that VM decomposition does a great job in rendering, which is exactly the novelty of TensoRF, and thus the paper misses quantitative comparisons with this representation, making the comparisons not convincing.

**Summary Of The Paper:**

The paper proposes TT-NF, a low-rank representation for learning neural fields, along with a sampling method for improve the training efficiency. It applies TT-NF to a synthetic tensor denoising task and a neural radiance field rendering task, which demonstrate the performance of the approach.

**Summary Of The Review:**

Although the paper has its novelty in applying TT to NF, my major concerns are the efficiency of TT (matrix multiplication vs point-wise multiplication) and its final performance, as discussed above. I think the current results are not convincing to me that TT-NF is superior than TensoRF.

---

> ### Author Response · Authors · 2022-11-16
> **Complexity of (Q)TT-NF**
>
> The complexity (FLOPs) of sampling from (Q)TT-NF is, in fact, $O(Dr^2)$, as there are no D matrix-matrix multiplications per each sample, but rather D matrix-vector multiplications. We acknowledge that [Chen et al. 2022] is linear in rank due to the different type of decomposition (triplanar). We do not see it as a reason to dismiss Tensor Train decomposition and QTT-NF in particular, even if a different decomposition suits the particular problem of learning the radiance field better.
> Regarding the complexity of producing a valid tensor train decomposition, our competitors are, e.g., TT-SVD, which produces results in a single iteration over cores, and TT-cross, which is an iterative algorithm and the only one before ours to recover TT from partial observation. Our algorithm is thus the first one after TT-cross to support partial observations, but on top of that, is resilient to noise and efficient (c.f. Figures 3, 4). Support for partial observations is important not just to enable stochasticity but also in cases when the full uncompressed tensor cannot be stored in the computer memory - a setting in which only TT-cross and TT-NF can perform.

---

> ### Author Response · Authors · 2022-11-16
> **Performance vs TensorRF**
>
> We clearly acknowledge that QTT-NF does not yet reach the performance of the full TensoRF. First, we do not see this as a ground for rejection. A monoculture where alternatives to the current state-of-the-art approach are no longer studied and explored would be extremely detrimental to the field. Second, in our paper, we focus on studying the Tensor Train parameterization of a neural field. Indeed, TensoRF is a “full package” that works well with a different (triplanar) type of decomposition and is the only work with a differentiable type of sparsity, which allows for direct comparison with our method when masking is disabled. For the same reason, we do not compare our method with Plenoxels, as they employ an uncompressed volume with pure masking, which is not a differentiable type of sparsity.
> Moreover, parameterizing radiance fields is just one of many potential applications of (Q)TT-NF. The focus of our paper is to study and advance the capabilities of the proposed framework. We do believe that (Q)TT-NF has great potential as a building block for future low-rank (hierarchical) representations and that it should be investigated further.

---

> ### Public Comment · ~Ivan_Oseledets1 · 2022-11-18
> **Significance**
>
> I don't agree that the main message of the paper the performance over TensorRF. TensorRF involves an unstable form of tensor decomposition (CP), which potentially may lead to problems for other datasets (CP decomposition and its rank in general is an NP-hard problem). For some reason, it was not mentioned at all in the original paper. It is an important problem.
> Tensor Train decomposition and methods for its usage are interesting on its own, and the baselines are clearly from the same type and shown that standard TT-sampling approaches can be significantly improved. This is also shown on the denoising example.
>
> A small comment is that the sampling is not cubic, it is quadratic.
>
> It is also an interesting observation in the paper that surprising efficiency of the TensorRT model (which also surprised me when I checked it) is due to the 'hidden' features in the code rather than the model itself. The paper does a great job in showing that.
> Finally, I don't think comparing to NERF is also important, since quite a lot of tensor decomposition baselines are considered.

---

### Official Review · Reviewer_2ByK · 2022-10-28

**Confidence:** 3
**Correctness:** 4
**Technical Novelty And Significance:** 3
**Empirical Novelty And Significance:** 2
**Recommendation:** 6

**Clarity, Quality, Novelty And Reproducibility:**

Per my comments above, I think that clarity could be improved, but the majority of the manuscript is reasonably clear. I have some reservations about novelty which I also expressed above. The results seem reproducible with the code provided in the supplement (I have not attempted to run it myself.)

**Strength And Weaknesses:**

The manuscript studies an important and very hot topic through the lens of efficient low-rank tensor decompositions. The authors' implementation indeed exhibits better robustness to noise than earlier decomposition algorithms, while offering more flexibility. The idea to use automatic differentiation in this task seems like the right thing to do, and the overall idea to use structured tensors to represent natural images and scenes is intriguing.

On the critical side, I am uncertain about the level of contribution. The idea to use structured tensors for radiance fields and novel view synthesis has been proposed before. The same is true for the tensor train ideas. Solving non-convex factorization problems by gradient-based optimization is also by now standard. I feel that many relevant comparisons are missing (please see below) and in the one with TensoRF they "cripple" the baseline to only compare the tensor-related algorithms. I can see why one may do that but it makes it unclear in what situation one should opportunely use QTT-NF. Some related questions below.

### Various questions and comments

- I am a bit confused by the first paragraph in the introduction: "Since, in extreme cases, the dimensionality of such fields can exceed the memory size of a typical computer by several orders of magnitude, we look at the problem of learning such fields from the angle of stochastic methods."
I suppose you mean dimensionality in some raw representation? But this depends on the chosen resolution. Since the underlying objects are continuous the resolution can be chosen as high as one desires but without introducing new information, so the notion of dimensionality is relative (as you show, these fields can be compressed). Can you back this up by concrete examples from practice?

- In the second paragraph you write "Both methods operate under the assumption of noise-free data and are not guaranteed to output sufficiently good approximations in the presence of noise."---this statement implies that data is regular in some predetermined sense and low-rank tensors respect this regularity. "Noise" would then be any excursion from this model. But there exists a great variety of models for images and other data---can you give an intuitive explanation for why TT-NF or any other tensor decomposition method should be the right choice?

- On p4, first paragraph, you write that "... $\cdot$ denotes merged (flattened) dimensions" but there are no $\cdot$s in the hierarchical expression a few lines up.

- What exactly does it mean that some algorithms "don't support noise in observations"? What happens when there _is_ noise? In particular, since this work is purely algorithmic, how would non-sub-gaussianity of noise manifest with TT-OI? (From Figure 4 it seems that TT-OI does a bit better than TT-SVD which according to your classification should not "support" noise.)

- is there a principled way to choose R_max in practice? is it possible to dynamically increase rank (e.g., if the loss stops decreasing at a value that is too high?)

- what is the influence of initialization (2) on the final result?

- This is to some degree semantics, but a lot of (most?) recent work on neural fields is about representations that can be queried at a continuum of input coordinates. There are many good reasons this is convenient, one important one is that it is easy to interface them with the various PDE solvers which evaluate spatial derivatives (often adaptively). Unless I am missing something, this is not possible with TT-NF. So TT-NF are primarily useful for compression, or some sort of regularization via the implicit multiscale inductive bias (from structured low-rankness).

- I find that the prose lacks clarity. A lot of it is too wordy (e.g. "... in the presence of access to full tensor elements...") and I had great difficulties following certain parts of the technical exposition. A case in point is the description of "v2" sampling scheme. (Up to this point I understood what is going on.) What does it mean to produce a batch of intermediates v? In what sense do linear layers require the samples to be packed densely? What is the alternative? Does overline over 2, D signify a range set? The statement "... we split the inputs v according to which M_i slices...". (After some effort I think I understand this part, but it should be written in a way that is friendlier to readers who only occassionaly dabble in tensor decompositions.)

- In Section 5.2, I understand the motivation to turn off the various techniques TensoRF uses to improve results, but then I find it a bit unfair to criticise it for grid alignment artifacts, at least without proper hedging. The method was designed as a package with a reason. For a fully transparent comparison I would like to see results for TensoRF with all the bells and whistles turned on, as well as results where neither TensoRF nor QTT-NF use the "tiny MLP".

- Further, it would be good to see comparisons with NERF or some state of the art neural-network based neural field on the rendering task. For full transparency, I would be interested in times (in addition to flops) required to fit some good neural model and QTT-NF on this task. I would also like to see comparisons like in Figure 3 for TT-{SVD, OI, cross}, perhaps also with the timings.

- Related to the above, I find it strange to use the term neural field for a tensor-based method which does not use a neural network.


**Summary Of The Paper:**

The authors propose a new way to sample tensor trains via efficient linear layers in deep learning frameworks. By further leveraging automatic differentiation, they propose to fit tensor trains to data by stochastic gradient descent and backpropagation. This gives a method that is robust to noise and that can be sampled at arbitrary batches of indices. The authors apply their sampling and fitting ideas to tensor denoising where it outperforms existing tensor decomposition methods. They further experiment with novel view synthesis and show improvements over a "bare-metal" version of a recent tensor based framework.


**Summary Of The Review:**

I think this is a nice paper (modulo a couple of editorial remarks) and the premise of using structured tensors to represent images and scenes is exciting. But currently I also feel that the  novelty is not quite at the ICLR level. This is compounded by some lack of transparency in the numerics.

## post-rebuttal (copy)

Thank you for the responses to my critical remarks.

About the first one: I don't think that suggesting that any of the reviewers advocate for a "monoculture" is a great move.

The comments of Ivan and Rafael adopt a better strategy which is to clarify the contributions of your paper. A good paper needs good results and good presentation. After reading your responses and Ivan's and Rafael's comments, I do think that I misjudged the novelty of your contribution. (Although I still think it is specialist and not at the level of a clear accept.)

Regarding the presentation, it is certainly possible that all reviewers made comments about neural fields, initializations, experiments, ..., because they are advocates of monocultures. But it could also be because the way you structured the story puts too much emphasis on these themes. Your TL;DR is not "We remove important bottlenecks in tensor decomposition algorithms by doing this and that" but "We repurpose the tensor train decomposition to learning compressed neural fields via backpropagation through samples." The abstract and the introduction share this emphasis.

Since I do feel that I mischaracterized the novelty (thanks, again, to the remarks of Ivan and Rafael) I will increase my score to a 6. But I will also reiterate my proposal that before employing lofty high-road arguments like "if you have it your way machine learning will be a monoculture!", the authors may consider the possibility that a different way to tell the story might have led to different discussions.

With regard to this last point, I don't think that a revised manuscript has yet been uploaded. I hope that the authors will consider restructuring the story to better emphasize the contributions if the paper is accepted.

---

> ### Author Response · Authors · 2022-11-16
> **Conclusion**
>
> We thank the reviewer for their time and effort in reading our submission and writing the review. Hopefully, our response addresses the main concerns stated in the review. We will implement the discussed improvements in the next revision of the paper. We urge the reviewer to evaluate our feedback and consider increasing their score.

---

> ### Author Response · Authors · 2022-11-16
> **Clarity of text and clarifications around v2 sampling**
>
> We thank the reviewer for pointing out these shortcomings; we will do our best to update the manuscript accordingly. Please find less formal extra clarifications below.
>
> A ‘batch of intermediates’ resulting from the first step of sampling algorithms refers to indexing the first TT-core of shape $1 \times M_1 \times R_1$ $B$ times using the first (out of $D$) coordinate. These $B$ first coordinates are part of the sampling algorithm input. The result of such slicing is a tensor of shape $1 \times B \times R_1$, which we reorder into a batch of matrices of shape $B \times 1 \times R_1$.
>
> Repeating such slicing using other batches of coordinates for cores 2 to $D$ ($i=\overline{2,D}$) results in tensors of shapes $B \times R_i \times R_{i+1}$. Storing these $D-1$ ‘intermediates’ and applying sequential batched-matrix-matrix multiplication to them to obtain the output of shape $B \times R_D$ is expensive and has a cubic complexity in $R_{max}$ (termed v1 sampling).
>
> To avoid replication of core parameters $B$ times for each sample, we intend to use linear layers. To do that, we look at the second group of $B$ coordinates along the dimension $M_2$ and want to apply matrix multiplication of slices of the second core with the corresponding elements of the batch of intermediates from the previous step. This would amount to separating the batch of intermediates into $M_2$ groups and applying slices of the second core as parameters of the linear layer to the respective groups. However, a linear layer’s input must be a matrix, and there is no guarantee that elements of the batch of intermediates corresponding to each slice of the second core are grouped together. To enforce such a grouping, we compute permutations of the batch of intermediates elements to group and align them with the respective slices of the second core. These steps correspond to Alg. 2 of the paper.

---

> ### Author Response · Authors · 2022-11-16
> **Continuum of input coordinates**
>
> Indeed, the representation is discrete, but this is well-aligned with both neural rendering and with the literature on tensor decompositions in continuous domains. Indeed, in neural rendering, we follow the convention of Plenoxels (also used in TensoRF) and perform trilinear interpolation of values from the eight nearest neighbors to obtain a continuous field. Recently, TT has been used to perform optimization of continuous parameter values in the reinforcement learning setting (NeurIPS’22 “TT-opt” paper), which could be discretized to sufficiently small steps thanks to mode quantization.

---

> ### Author Response · Authors · 2022-11-16
> **Influence of initialization on the final result**
>
> Random initialization, as in Eq. (2) of the paper, is recommended to match the first and second moments of the distribution of uncompressed tensor elements to the standard gaussian. This initialization works sufficiently well in both Tensor Denoising and Neural Rendering settings. For the former, this is due to the prior knowledge of the task - by construction, the ground truth is obtained using the same principle, so initialization with the same statistical properties leads to fast convergence, which is a motive exploited in nearly every non-convex optimization setting. For the latter (Neural Rendering), we exploited a training protocol assumed in training NeRF with a neural network. Most neural network initialization approaches preserve zero mean and unit variance of the network outputs. Thus, with such an initialization, one can directly plug in (Q)TT-NF in the place of coordinate-based neural networks.
>
> While neural networks allow initialization with pre-trained parameters, (Q)TT-NF supports that too, but additionally provides a completely new way to directly initialize parameters with the values via the TT decomposition of the uncompressed tensor (obtained with any other method of choice, e.g., from Table 1). Experiments with instantaneous initialization from TT-SVD and fine-tuning QTT-NF are provided in Table 3 of the paper (Supplementary materials section).

---

> ### Author Response · Authors · 2022-11-16
> **Principled way to choose R_max**
>
> As with most empirical best practices in ML, a reliable way would require choosing a sufficiently small R_max, fitting it on the “training” split, evaluating fitting error on the “validation” split, and checking that against some stopping criterion. This is how the TT-cross algorithm works, too. Observing the chosen loss values is also possible, but we did not explore it as a stopping criterion.
>
> With TT (and (Q)TT-NF by extension), it can increase rank dynamically. To do that, one should pass a new list with TT-rank values $(1, R_1^\mathrm{(new)}, …, R_{D-1}^\mathrm{(new)}, R_D)$, where $R_i^\mathrm{(new)} \geq R_i^\mathrm{(old)}, i=\overline{1,D-1}$, and pad the existing TT-cores with zeros along the corresponding rank dimensions. After that, these new padded cores can be used to reinitialize learned parameters of (Q)TT-NF and continue learning from exactly the same representation, yet higher capacity.
>
> Moreover, (Q)TT-NF can scale up spatial capacity by multiplicative factors by introducing a new TT-core before the last one, corresponding to the payload. This new core needs to be initialized with stacked identity matrices. Such an operation would produce the same result as if the uncompressed tensor was nearest-neighbor upsampled.
>
> We hope to have illustrated the great flexibility of the novel learned representation we propose.

---

> ### Author Response · Authors · 2022-11-16
> **Algorithmic support for noise in observations**
>
> For non-Gaussian noise in observations, c.f. Figure 4 of the paper; the right set of plots corresponds to Laplacian noise added to the ground truth. As can be seen, TT-OI results fluctuate and have a higher error than the baseline (TT-SVD). Overall, we found TT-OI not very resilient (not even against Gaussian noise when the underlying rank is higher than r=2, the value reported in Fig. 5 of the original paper).
>
> Formally, “support” for noise by the algorithm means that it is designed to output a representation with a certain theoretically-derived worst-case bound. That bound is not the same as the empirical average behavior - for instance, TT-SVD outputs a decent approximation. Empirically it is still worse than the one obtained through stochastic optimization of TT-NF, as seen in all plots except for the case in the bottom-left corner (corresponding to little normal noise and a rather high underlying rank).
>
> These observations underline the practical utility of our proposed learned representation.

---

> ### Author Response · Authors · 2022-11-16
> **Intuition for using tensor decompositions for modeling data**
>
> The intuition comes from the low-rank nature of visual data. That is, sensible visual data form a low-dimensional manifold in the data space. For example, QTT-NF applied to compress 2D images exhibits the same artifacts as JPEG DCT quantization. In the field of 3D data modeling, tensor decompositions have been used even before deep learning (c.f. “Tensor Decompositions in 3D Representations” paragraph of the “Related Work” section). The TT decomposition is a type of sparsity that can be used with any type of data but also one that suits visual data well, as shown by earlier literature. Our work provides a computational scheme to fit the TT representation to noisy observations, addressing two main limitations of previous methods (c.f. Table 1 in the paper).

---

> ### Author Response · Authors · 2022-11-16
> **Confusion about “extreme cases” and their dimensionality in the Introduction**
>
> We used the term ‘dimensionality’ loosely here, referring to the number of elements in an uncompressed tensor, which obviously grows exponentially with the number of tensor dimensions. Increasing resolution is a way to go as long as one can store it or compute elements on the fly. This is the point we are making: raw storage becomes intractable already at rather moderate resolutions, regardless of the content. We will clarify this in the text.

---

> ### Author Response · Authors · 2022-11-16
> **QTT-NF vs TensoRF**
>
> We clearly acknowledge that QTT-NF does not yet reach the performance of the full TensoRF. First, we do not see this as a ground for rejection. A monoculture where alternatives to the current state-of-the-art approach are no longer studied and explored would be extremely detrimental to the field.
>
> Second, in our paper, we focus on studying the Tensor Train parameterization of a neural field. Indeed, TensoRF is a “full package” that works well with a different (triplanar) type of decomposition and is the only work with a differentiable type of sparsity on neural rendering, which allows for direct comparison with our method when masking is disabled. For the same reason, we do not compare our method with Plenoxels, as they employ an uncompressed volume with pure masking, which is not a differentiable type of sparsity.
>
> Moreover, parameterizing radiance fields is just one of many potential applications of (Q)TT-NF. The focus of our paper is to study and advance the capabilities of the proposed framework. We do believe that (Q)TT-NF has great potential as a building block for future low-rank (hierarchical) representations and that it should be investigated further.

---

> ### Author Response · Authors · 2022-11-16
> **Novelty**
>
> We respectfully disagree with the reviewer’s claim about weak novelty. To the best of our knowledge, our work is the first to show how to update the TT parameterization via gradient-based optimization efficiently in the stochastic setting. That is, we introduce a novel, learned representation and algorithms to work with it, which perfectly aligns with the scope of ICLR. If the reviewer is aware of any prior work that does this, we would be happy if they could point us to a concrete reference.

---

### Public Comment · ~Rafael_Ballester-Ripoll1 · 2022-11-17
**Comment on the paper novelty w.r.t. tensor decompositions**

I would like to weigh in regarding the paper's merits in the tensor decomposition front. The proposed octet representation that merges x, y, z subindices level-wise is a promising new improvement over standard QTT, and could act as a drop-in replacement in other applications where QTT is beneficial (there are many). A similar point can be made about the sampling algorithm: batched reconstruction from tensors in the TT format (and variations thereof) is a very common bottleneck in learning pipelines that exploit the low-rank ansatz, and the proposed scheme to alleviate the computational burden is certainly welcome. Last, the ability to learn noisy tensors is valuable per se, since the much celebrated cross-approximation algorithm has been shown to be unstable for this task. Indeed, the scheme is more flexible than cross-approximation in the sense that it can be used when the set of available samples is dictated by an external process (ray traversal paths in this case, but there can be many other scenarios) and cannot be acquired on demand as the cross method requires. In the context of learning tensor decompositions from given data, this is a novel contribution in its own right.

---

### Public Comment · ~Ivan_Oseledets1 · 2022-11-18
**Novelty**

I've added a couple of answers to the reviewers, just to summarize: I think this paper provides novelty for tensor decomposition field, and solves an important problem in the field.

---

### Decision · Program_Chairs · 2023-01-20

**Decision:**

Reject

**Justification For Why Not Higher Score:**

All scores clearly below threshold

**Justification For Why Not Lower Score:**

N/A

**Metareview: Summary, Strengths And Weaknesses:**

In this paper, the authors introduce a method called Tensor Train Neural Fields (TT-NF) for learning neural fields on dense regular grids. They use backpropagation to train the TT-NF representation, which is a low-rank parameterization of the neural field, and minimize a non-convex objective. They show various experiments by applying it to a tensor denoising task and Neural Radiance Fields. The reviewers thought the paper studies an important and contemporary topic through the lens of efficient low-rank tensor decompositions and thought the robustness to noise experiments were interesting. The reviews however raised concerns about the novelty of using tensors for radiance fields and novel view synthesis given prior work in the area, novelty w.r.t. tensor train methods, and lack of comparison with many baselines, clarity of writing, and the need for many multiplications. One reviewer also thought "the experimental results on Synthetic NeRF dataset is not convincing". While some of the concerns were alleviated based on the authors thorough response many were not. Based on my own reading this paper is interesting and has potential but is also clear that the above issues need to be properly addressed before the paper can be accepted for publication. I therefore recommend rejection.